# Exploration of efficient on-device acoustic modeling with neural networks

## Abstract

Real-time speech recognition on mobile and embedded devices is an important application of neural networks. Acoustic modeling is the fundamental part of speech recognition and is usually implemented with long short-term memory (LSTM)-based recurrent neural networks (RNNs). However, the single thread execution of an LSTM RNN is extremely slow in most embedded devices because the algorithm needs to fetch a large number of parameters from the DRAM for computing each output sample. We explore a few acoustic modeling algorithms that can be executed very efficiently on embedded devices. These algorithms reduce the overhead of memory accesses using multi-timestep parallelization that computes multiple output samples at a time by reading the parameters only once from the DRAM. The algorithms considered are the quasi RNNs (QRNNs), Gated ConvNets, and diagonalized LSTMs. In addition, we explore neural networks that equip one-dimensional (1-D) convolution at each layer of these algorithms, and by which can obtain a very large performance increase in the QRNNs and Gated ConvNets. The experiments were conducted using two tasks, one is the connectionist temporal classification (CTC)-based end-to-end speech recognition on WSJ corpus and the other is the phoneme classification on TIMIT dataset. We not only significantly increase the execution speed but also obtain a much higher accuracy, compared to LSTM RNN-based modeling. Thus, this work can be applicable not only to embedded system-based implementations but also to server-based ones.

## 1 Introduction

Acoustic modeling is the fundamental part of automatic speech recognition, and is a representative application of recurrent neural networks (RNNs) (Graves et al., 2013; Amodei et al., 2016; Miao et al., 2015). The sequence learning algorithms are usually executed sequentially in the time domain to utilize the result of the previous timesteps. For example, in LSTM RNNs, the output of the previous time step, $\mathbf{h}_{t-1}$, is used for computing the gate signals of the current timestep, $t$ (Hochreiter & Schmidhuber, 1997). Thus, we need to compute $\mathbf{h}_{t-1}$, $\mathbf{h}_t$, and $\mathbf{h}_{t+1}$ sequentially. This is extremely disadvantageous when implementing the algorithms on embedded devices because the network parameters must be retrieved from the external DRAM at every timestep. The execution of an LSTM RNN on a GPU does not suffer from this problem because multiple LSTM RNNs are executed simultaneously while sharing the network parameters among the RNNs. However, in embedded devices, the algorithm is usually executed as a single stream, and has no way of sharing the parameters with other sequences. One conventional solution to overcome this problem was reducing the size of the parameters by quantization, pruning, and data compression (Narang et al., 2017; Xu et al., 2018; Prabhavalkar et al., 2016). However, the size of the parameters can be too big to accommodate on internal memory or cache of embedded systems.

A model of the hardware of the embedded systems for executing the acoustic model is shown in Figure 1. The hardware is based on the current generation of ARM Cortex-A57 CPU that contains four cores and L2 cache memory of 2 MB. The end-to-end acoustic model considered in the research contains at least 10 million parameters, which can reach 40 MB when implemented as a floating-point. We consider that the efficient execution of neural networks, with parameter sizes larger than that of the cache or internal memory capacity, is a very important research topic. This work is an algorithmic level optimization approach, and the multi-timestep parallel technique exploited in this research can be combined with the quantization and pruning techniques.

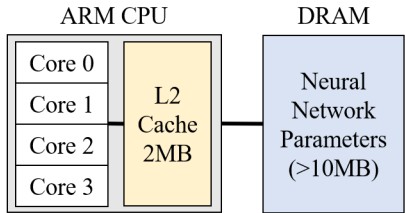

Figure 1: Embedded hardware model for executing neural networks.

Recently, quasi RNNs (QRNNs) and simple recurrent units (SRUs) were developed for the purpose of fast training and inference of very long sequences (Balduzzi & Ghifary, 2016; Bradbury et al., 2017; Lei et al., 2017; Martin & Cundy, 2018). These RNNs only employ simple feedback that can be represented by linear recurrence equations, and allow fast training as well as multi-timestep parallel inference. QRNNs have a good performance comparable to LSTM in language modeling and machine translation. Convolutional neural networks (CNNs) have also been used for sequence learning, such as machine translation, language modeling, and speech recognition (Gehring et al., 2017; Dauphin et al., 2017; Liptchinsky et al., 2017; Kang et al., 2017). A CNN has the advantage of parallel processing because there is no dependency between the input and output. However, CNNs only provide a finite context length, while RNN can consult infinite length, at least in theory. Besides these networks, a few different RNN structures have been devised in recent years, and they can be utilized for efficient implementation in embedded systems (Subakan & Smaragdis, 2017; Li et al., 2018). However, when applied to acoustic modeling, the accuracy of these sequence-modeling algorithms has not been as good as that of LSTM RNNs.

In this work, we assess the performance of several sequence modeling algorithms including diagonal LSTM, QRNN, and Gated ConvNet, and compare them with LSTM RNN. The performance was measured using CTC-based end-to-end acoustic modeling and frame-level phoneme recognition tasks. Our initial experiments showed that LSTM RNN performed best, thus we modified the networks by including 1-D convolution to each layer. Note that adding 1-D convolution can increase the length of the context. We find several non-LSTM-based algorithms that yield a much better performance when compared to LSTM RNN. We also vary the depth and width to improve the performance. The execution performances of these algorithms on embedded devices are also compared.

This paper is organized as follows. In Section 2, we introduce a few acoustic modeling algorithms that permit multi-timestep parallelization. Section 3 provides the experimental results of the acoustic models on end-to-end speech recognition and framewise phoneme classification tasks. Section 4 discusses related studies. The concluding remarks are shown in Section 5.

## 2 ACOUSTIC MODELING ALGORITHMS

The most well-known and widely used sequence-learning algorithm is the LSTM RNN, represented by Eq. 1. As illustrated in this equation, the algorithm employs two feedback structures; one is an element-wise feedback for updating $\mathbf{c}_t$, and the other is the complex feedback for computing $\mathbf{f}_t$, $\mathbf{i}_t$, and $\mathbf{o}_t$ using the previous output $\mathbf{h}_{t-1}$. Here, we assume that the total sizes of the matrices $\mathbf{U}_f$, $\mathbf{U}_i$, $\mathbf{U}_o$ and $\mathbf{U}_c$ are larger than that of the cache memory in embedded systems. Thus, when executing the LSTM RNN on embedded devices, the parameters must be retrieved at every inference, which incurs large memory access overhead.

$$
\begin{aligned}
\mathbf{f}_t &= \sigma(\mathbf{W}_f \mathbf{x}_t + \mathbf{U}_f \mathbf{h}_{t-1} + \mathbf{b}_f), \\
\mathbf{i}_t &= \sigma(\mathbf{W}_i \mathbf{x}_t + \mathbf{U}_i \mathbf{h}_{t-1} + \mathbf{b}_i), \\
\mathbf{o}_t &= \sigma(\mathbf{W}_o \mathbf{x}_t + \mathbf{U}_o \mathbf{h}_{t-1} + \mathbf{b}_o), \\
\mathbf{c}_t &= \mathbf{f}_t \odot \mathbf{c}_{t-1} + \mathbf{i}_t \odot \tanh\left(\mathbf{W}_c \mathbf{x}_t + \mathbf{U}_c \mathbf{h}_{t-1} + \mathbf{b}_c\right), \\
\mathbf{h}_t &= \mathbf{o}_t \odot \tanh(\mathbf{c}_t).
\end{aligned}
\tag{1}
$$

To solve this problem, we explore three algorithms. The first is the diagonal LSTM that employs simple diagonal matrices for $\mathbf{U}_f$, $\mathbf{U}_i$, $\mathbf{U}_o$, and $\mathbf{U}_c$; the second is the QRNN that does not use $\mathbf{U}_f$, $\mathbf{U}_i$, $\mathbf{U}_o$ and $\mathbf{U}_c$; and the third is the Gated ConvNet that does not employ any recurrent loops.

## 2.1 DIAGONAL LSTM RNN

Matrix-vector multiplication is widely used in neural networks, however, it demands a large number of parameters as well as arithmetic operations. Thus, element-wise multiplication operations are quite advantageous, and one example is the computation of $\mathbf{c}_t$ in Eq. 1. The element-wise multiplication is equivalent to the multiplication of a diagonal matrix and a vector.

The diagonal LSTM RNN only allows the diagonal terms of $\mathbf{U}_f$, $\mathbf{U}_i$, $\mathbf{U}_o$ and $\mathbf{U}_c$ to be non-zero in the LSTM equation (Subakan & Smaragdis, 2017; Li et al., 2018). In embedded implementations, we can assume that the diagonal terms can be stored in the cache memory because their size is much smaller than that of the full matrices. The number of words for the diagonal terms is a few thousands, while that of the full matrices is millions. This architecture was used for music sequence prediction, and showed a good performance (Subakan & Smaragdis, 2017). However, the efficacy of this algorithm for acoustic modeling has not been studied.

When implementing this algorithm, we compute $[\mathbf{W}_f, \mathbf{W}_i, \mathbf{W}_o, \mathbf{W}_c] \times \mathbf{x}_t$, for $t = 1, 2, \ldots, T_P$, at one time, which can be completed by fetching the parameters only once from the DRAM. Thus, the overhead of parameter access can be reduced in proportion to $T_P$. Note that the rest of the procedure for computing $\mathbf{h}_t$, for $t = 1, 2, \ldots, T_P$, can be conducted without accessing the external DRAM, particularly because the sizes of $\mathbf{U}_f$, $\mathbf{U}_i$, $\mathbf{U}_o$, and $\mathbf{U}_c$ are quite small due to diagonalization.

## 2.2 QRNN

The element-wise operations, or the multiplication of a diagonal matrix and a vector, are restricted compared to the matrix-vector multiplications; the former should assume the same basis for the operand and the result, while the latter has the ability of altering the basis. The computation of $\mathbf{c}_t$ in Eq. 1 uses $\mathbf{c}_{t-1}$ as the operand, thus element-wise operations can work. Simplification of the LSTM RNN using element-wise operations is currently of great interest (Balduzzi & Ghifary, 2016; Bradbury et al., 2017; Lei et al., 2017). The simplest form of QRNN is shown in Eq. 2.

$$\begin{aligned}
\hat{\mathbf{x}}_t &= \tanh\left(\mathbf{W}_z \mathbf{x}_t + \mathbf{b}_z\right), \\
[\mathbf{f}_t, \mathbf{i}_t, \mathbf{o}_t] &= \sigma([\mathbf{W}_f, \mathbf{W}_i, \mathbf{W}_o]\mathbf{x}_t + [\mathbf{b}_f, \mathbf{b}_i, \mathbf{b}_o]), \\
\mathbf{c}_t &= \mathbf{f}_t \odot \mathbf{c}_{t-1} + \mathbf{i}_t \odot \hat{\mathbf{x}}_t, \\
\mathbf{h}_t &= \mathbf{o}_t \odot \mathbf{c}_t + (1 - \mathbf{o}_t) \odot \mathbf{x}_t.
\end{aligned} \tag{2}$$

QRNN does not employ the feedback with $\mathbf{h}_{t-1}$, thus it does not use $\mathbf{U}_f$, $\mathbf{U}_i$, $\mathbf{U}_o$, and $\mathbf{U}_c$. Instead, this algorithm can consult $k$ input samples, $\mathbf{x}_t, \mathbf{x}_{t-1}, \ldots, \mathbf{x}_{t-k+1}$, while LSTM uses only $\mathbf{x}_t$. The QRNN equation when $k = 1$ is represented by Eq. 2. The parameter size of QRNN is linearly dependent on $k$, thus it is not advantageous to use a large value of $k$. The simplest case, $k = 1$, is called the simple recurrent unit (SRU) (Lei et al., 2017).

## 2.3 GATED CONVNET

A CNN does not employ any recurrent loops, but it can conduct sequence modeling of finite length. Although this architecture should assume a finite context length, the modeling can solve many practical problems, such as language and acoustic modeling. Although CNNs may have a limited context length due to the feed-forward only structure, the architecture is relatively free from the gradient exploding problems. One variant of CNN-based architecture is the Gated ConvNet, which has a good performance in acoustic and language modeling (Dauphin et al., 2017; Liptchinsky et al., 2017). The equation for the Gated ConvNet with the filter length of $T$ is shown in Eq. 3,

$$\mathbf{h}(\mathbf{X}) = (\mathbf{X} * \mathbf{W} + \mathbf{b}) \odot \sigma(\mathbf{X} * \mathbf{V} + \mathbf{c}) \tag{3}$$

where $*$ denotes convolution operation and $\odot$ represents element-wise multiplication. $\mathbf{W}, \mathbf{V} \in \mathbb{R}^{T \times w \times w}$ and $\mathbf{b}, \mathbf{c} \in \mathbb{R}^w$ are trainable variables where $w$ is the width of the network. $\mathbf{X} \in \mathbb{R}^{N \times w}$ is the input vector where $N$ is the sequence length.

The parameter size of the Gated ConvNet increases proportional to the filter length $T$ and the depth $D$, while the context length corresponds to $D \times T$. Thus, there is compromise between the parameter size and the context length. The Gated ConvNet structure can be considered to employ 2-dimensional (2-D) convolutions in multilayers.

## 2.4 Element wise 1-D convolution

Increasing the context length in the time domain is critical for an improved performance. However, the parameter size increases proportional to the context length. When the filter length is $T$ and the width of the neural network is $w$ for both the input and output, the parameter size is proportional to $T \times w^2$. One way of reducing the parameter size and increasing the context length is to employ element-wise time domain convolution, which is called 1-D convolution in this work. In this case, the parameter size increase is negligible due to dimensionality reduction. However, this assumes that the feature map elements are independent, and consulting other elements is not beneficial.

We conducted experiments by adding the 1-D convolution to the QRNN and Gated ConvNet, and a minimum value of the 2-D filter length $T$ was employed for the sake of reducing the parameter sizes. The 1-D convolution can be causal or consult the future input. We denote the 1-D convolution coefficients as $\mathbf{h}_t$. When $t$ is a negative value, it denotes consulting the future input. The 1-D parameter of $\mathbf{h}_0, \ldots, \mathbf{h}_4$ means that it consults the past four inputs including the current one.

# 3 Experimental results

We conducted experiments with two acoustic modeling tasks. One is the CTC algorithm-based end-to-end speech recognition that emits characters, word-pieces, or words directly from the acoustic input (Graves et al., 2013). Since the output is text, the modeling itself is speech recognition, although language modeling-based post-processing is usually employed for increased accuracy. This task is considered a very difficult sequence modeling, and LSTM-based RNNs have been used in most of the implementations. The Wall Street Journal (WSJ) corpus (Paul & Baker, 1992) was used for this experiment. The other task used for the experiment is the frame-wise phoneme classification; the phoneme label is provided for each frame at the training time (Graves & Schmidhuber, 2005; Hinton et al., 2012). Thus, this task can be considered a pattern classification problem, and simple multilayer perceptron and CNN can be employed. However, since the acoustic signal is highly predictable, the LSTM RNN has demonstrated a much better performance than MLP or CNN. The TIMIT dataset is used for this phoneme classification test.

## 3.1 End-to-end speech recognition

The experiments were conducted with WSJ SI-284, which contains 81 hours of speech, for fast exploration on 4x600 LSTM, 6x800 QRNN with $k = 1$, 6x700 diagonal LSTM, and 6x300 Gated ConvNets with the context length $T$ of 7 and 15. When modeling a network, the first number represents the depth, while the second is the width. For example, 4x600 LSTM refers to the network with a depth of 4 and a width of 600.

The overall acoustic modeling architecture is shown in Fig. 2 (Sainath et al., 2015; Amodei et al., 2016). It comprises two 2-D CNN layers, multiple neural network layers, such as LSTM, QRNN, or ConvNet, and the final fully connected layer. The output labels are graphemes (characters), and the AM is trained with CTC loss. The input of the convolutional layer consists of three 2-D feature maps with the time and frequency axes; each feature map is formed with the Mel-filter bank output, its delta, and double-delta. The 2-D convolutional layers employ a filter size of 5, as proposed in (Amodei et al., 2016). The 2-D CNN layers not only improve the recognition performance but also help reduce the arithmetic complexity by down-sampling the input frames by two.

A 40-dimensional log Mel-frequency filter-bank feature is extracted from the raw speech data. The feature vectors are sampled every 10 ms with a 25 ms Hamming window. We applied batch normalization to the first two convolutional layers and variational dropout to every output of the recurrent

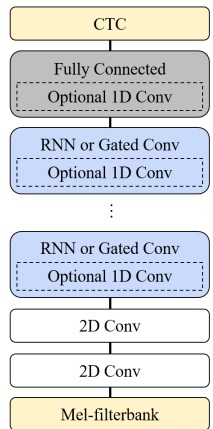

Figure 2: Acoustic modeling architecture for end to end speech recognition.

layer for regularization (Ioffe & Szegedy, 2015; Gal & Ghahramani, 2016). Adam optimization was applied for training (Kingma & Ba, 2014). We used an initial learning rate of 3e-4, which was reduced to half if the validation error was not lowered for 8 consecutive epochs. Gradient clipping with a maximum norm of 4.0 was applied if needed. For comparison, we trained all the models with an identical hyper-parameter setting. The trained models were evaluated on WSJ eval92 set.

Table 1 shows the model size, character error rate (CER), and word error rate (WER) of the models trained with the WSJ SI-284. The greedy decoding results are shown, which do not conduct post-processing with language models. This table shows that LSTM RNN achieves the best performance with a WER of 24.88%, and the QRNN shows the worst performance among these. This suggests that the local feedback with $\mathbf{h}_{t-1}$ is essential for achieving a good performance. Note that QRNN does not use feedback from the previous output, $\mathbf{h}_{t-1}$.

Table 1: WER and CER (%) evaluated on WSJ eval92. The models are trained on SI-284.

| Models | Params. | CER | WER |
|---|---|---|---|
| 4x600 LSTM | 11.5M | **7.29** | **24.88** |
| 6x800 QRNN ($k = 1$) | 11.5M | 12.70 | 45.22 |
| 6x700 Diagnoal LSTM | 11.5M | 8.97 | 30.40 |
| 6x300 Gated ConvNet ($T = 15$) | 16.2M | 8.02 | 28.65 |
| 6x300 Gated ConvNet ($T = 7$) | 7.7M | 8.52 | 30.56 |

As for the subsequent experiments, we added a 1-D convolution to each layer of neural networks. The 1-D convolution has a time-length of 15, from -7 to +7. It is designed to consult the future input. Since the number of parameters for 1-D convolution is very small, the model size is almost unaffected. The experimental results with 1-D convolution are shown in Table 2. As shown in this table, the performance of LSTM showed almost no change, with a small WER value drop from 24.88% to 23.57%. However, diagonal LSTM, QRNN, and Gated ConvNet all displayed quite good performance gain. The performance of LSTM with 1-D convolution is almost same as that of diagonal LSTM with 1-D convolution. Hence, the diagonal LSTM with 1-D convolution is much more advantageous when compared to conventional LSTM because it allows multi-timestep processing in embedded systems. The best performance was obtained with QRNN with 1-D convolution. It achieved a WER of 19.07%, which is very impressive compared to that of LSTM, 24.88%. Gated ConvNets with $T = 2$ showed a fairly good performance even with a small number of parameters.

To improve the performance, we increased the depth of the networks. It is well known that increasing the depth helps to improve the performance in many cases. However, deeper networks are hard to

Table 2: WER and CER (%) evaluated on WSJ eval92 with 1-D convolution.

| Models | Params. | CER(%) | WER(%) |
|---|---|---|---|
| 4x600 LSTM, 1-D conv | 11.5M | 6.95 | 23.57 |
| 6x700 QRNN ($k = 1$), 1-D conv | 11.5M | 5.26 | 19.07 |
| 6x700 Diagonal LSTM, 1-D conv | 11.5M | 7.57 | 23.90 |
| 6x300 Gated ConvNet ($T = 15$), 1-D conv | 16.2M | 7.58 | 27.00 |
| 6x300 Gated ConvNet ($T = 7$), 1-D conv | 7.7M | 6.57 | 24.20 |
| 20x300 Gated ConvNet ($T = 2$), 1-D conv | 7.5M | 5.55 | 19.70 |
| 30x300 Gated ConvNet ($T = 2$), 1-D conv | 11M | **4.73** | **17.00** |

train, especially for LSTM RNN. For the Gated ConvNet, residual connections were employed to help the gradient flow smoothly through the deep networks (He et al., 2016). Table 2 also shows the experimental results with 15- and 30-layer Gated ConvNets. The 30-layer Gated ConvNet showed a greedy decoding WER of 17.00% with WSJ SI-284, which is the quite remarkable result.

We also show the performance of end-to-end acoustic models with WSJ SI-ALL corpus, which contains 147 hours of speech. The decoding results with the character-level language model are shown in Table 3. The trained models were evaluated on WSJ eval92 set. The beam search decoding was conducted using the same Hierarchical Character Level Language Model (HCLM). The HCLM has four 512-dimensional LSTM layers, of which two layers are assigned to the word-level modeling (Hwang & Sung, 2017). RNN LM was trained on WSJ LM training text. We randomly selected 5% of WSJ LM training text to the valid set, and another 5% to the test set. The remaining 90% of the text was used for training RNN LM. The HCLM reported 1.07 of bit-per-character (bpc) on the test set. The decoding was conducted with a beam width of 128 (Hwang & Sung, 2016). The obtained WER after HCLM decoding is below 4%, which is almost comparable to DeepSpeech2 (Amodei et al., 2016) which showed the WER of 3.60% with 100M parameters.

Table 3: WER and CER (%) on WSJ eval92. The models are trained on WSJ SI-ALL.

| Models | Params. | Greedy | | HCLM | |
|---|---|---|---|---|---|
| | | CER | WER | CER | WER |
| 4x600 LSTM, 1-D conv | 11.5M | 5.91 | 20.14 | 2.71 | 6.56 |
| 6x700 QRNN ($k = 1$), 1-D conv | 11.5M | 4.13 | 18.02 | **1.51** | **3.73** |
| 30x300 Gated ConvNet ($T = 2$), 1-D conv | 11M | **3.30** | **11.60** | 1.53 | 3.86 |
| Deep Speech 2 (Amodei et al., 2016) | 100M | WER 3.60 with 5-gram LM | | | |

## 3.2 PHONEME CLASSIFICATION

The phoneme classification with TIMIT dataset was conducted to verify the efficacy of 1-D time convolution in a different acoustic modeling task. This problem required much less training time compared to the end-to-end training with WSJ, and allowed us to know the effect of the different parameters, such as the 1-D filter length, within a short time.

Although phoneme classification can be conducted with MLP and CNN, LSTM RNN is efficient (Graves & Schmidhuber, 2005). We conducted experiments following this research, but the log Mel-frequency bank spectrum was used as the input features instead of the Mel-Frequency Cepstral Coefficients (MFCCs). For evaluation, the original 61 phones were folded into standard 39 classes as proposed in (Lee & Hon, 1989).

Table 4 shows the frame-wise correct phoneme classification rate of sequence models studied in this research. The parameter sizes are restricted to around 1 million to follow the experiments in (Graves & Schmidhuber, 2005). We consider that the models do not need to be big because the number of data are small, and the frame-wise phoneme classification is a simple task compared to end-to-end speech recognition.

Table 4: Frame-wise phoneme classification accuracy (%) on TIMIT.

| Models | Params. | Accruacy |
|---|---|---|
| 2x256 LSTM | 0.92M | 72.00 |
| 4x256 QRNN ($k = 1$) | 0.92M | 52.74 |
| 4x256 Diagonal LSTM | 0.92M | 68.45 |
| 4x256 Gated ConvNet ($T = 2$) | 0.91M | 62.00 |
| 2x256 LSTM, 1-D conv | 0.93M | 75.30 |
| 4x256 Diagonal LSTM, 1-D conv | 0.93M | 74.60 |
| 4x256 QRNN ($k = 1$), 1-D conv | 0.93M | 76.48 |
| 4x256 Gated ConvNet ($T = 2$), 1-D conv | 0.92M | 76.30 |
| 10x128 Gated ConvNet ($T = 2$), 1-D conv | 0.67M | **77.04** |

Table 4 shows the phoneme classification rate of about 72.00% with LSTM RNN, with a model configuration of 2x256. This result is better than the classification rate of 66.00% for LSTM reported in (Graves & Schmidhuber, 2005). The QRNN with a configuration of 4x256 only exhibits the accuracy of 52.74%, which is not a good performance. The Gated ConvNet showed the accuracy of 62%, which is not as good as that of LSTM. However, the performance of QRNN and Gated ConvNet with 1-D convolution showed very good accuracy: 76.57% for QRNN with 1-D and 77.04% for Gated ConvNet with 1-D convolution. LSTM with 1-D convolution did not show a significant performance improvement. All 1-D convolution employs the configuration of -5 to 5. We conducted experiments using different configurations with Gated ConvNet. Here, we can find that the depth is important in achieving a good performance. The accuracy obtained from 10x128 Gated ConvNet with 1-D convolution is the highest in this experiment. Table 5 shows the performance of various Gated ConvNets on the phoneme classification task.

Table 5: Frame-wise phoneme classification accuracy (%) of different Gated ConvNets on TIMIT.

| Models | Params | Accuracy |
|---|---|---|
| 4x256 Gated ConvNet ($T = 2$), 1D conv (-3,+3) | 0.92M | 76.44 |
| 4x256 Gated ConvNet ($T = 2$), 1D conv (-5,+5) | 0.92M | 76.30 |
| 4x256 Gated ConvNet ($T = 2$), 1D conv (-7,-7) | 0.92M | 73.90 |
| 10x128 Gated ConvNet ($T = 2$), 1D conv (-3,+3) | 0.67M | **78.04** |
| 10x128 Gated ConvNet ($T = 2$), 1D conv (-5,+5) | 0.67M | 77.04 |

## 3.3 IMPLEMENTATION RESULTS ON EMBEDDED SYSTEMS

The LSTM RNN computes the output, $\mathbf{h}_t$, one at a time, and, as a result, the number of DRAM accesses can be excessive because the parameters cannot be stored on the cache memory. In QRNN-based implementations, the $T_P$ output samples are computed at a time by retrieving one set of parameters. Therefore, the matrix-vector multiplication for multiple timesteps can be merged into a single matrix-matrix multiplication as follows:

$$\begin{bmatrix} \mathbf{z}_1 & \mathbf{z}_2 & ... & \mathbf{z}_{T_P} \\ \mathbf{i}_1 & \mathbf{i}_2 & ... & \mathbf{i}_{T_P} \\ \mathbf{f}_1 & \mathbf{f}_2 & ... & \mathbf{f}_{T_P} \\ \mathbf{o}_1 & \mathbf{o}_2 & ... & \mathbf{o}_{T_P} \end{bmatrix} = \begin{pmatrix} \mathbf{W}_z \\ \mathbf{W}_i \\ \mathbf{W}_f \\ \mathbf{W}_o \end{pmatrix} (\mathbf{x}_1 \; \mathbf{x}_2 \; ... \; \mathbf{x}_{T_P}) \tag{4}$$

where $T_P$ is the number of parallelization steps. The multi-timestep processing converts matrix-vector multiplications into a matrix-matrix multiplication. The weight matrix is reused for $T_P$ time steps with a single parameter from the DRAM, by which the execution time and power consumption can be greatly reduced. OpenBLAS library was used for the computation of Eq. 4 (Xianyi et al., 2012).

The remaining steps do not demand much computation because element-wise multiplication or vector addition are much simpler than matrix-vector multiplication. Since we computed the $T_P$ output samples with one set of parameters retrieved, the memory access overhead was reduced proportional to $T_P$. However, the delay grows as $T_P$ increased. Gated ConvNets do not have recurrent paths, thus multi-timestep parallelization can be easily applied.

Table 6 shows the execution time of end-to-end acoustic models implemented in this work. The execution time is measured for the multi-time parallel steps of 1 (sequential execution), 8, and 16. As can be seen in this table, the QRNN and Gated ConvNet based algorithms are 7.30 and 5.03 times faster than LSTM when the output for eight timesteps are computed at a time.

Table 6: Execution time (sec) for models for 1 sec speech, function of $T_P$

| Models | $T_P = 1$ | $T_P = 4$ | $T_P = 8$ | $T_P = 16$ |
|---|---|---|---|---|
| 4x600 LSTM | 1.46 | | | |
| 6x700 QRNN ($k = 1$), 1-D conv | 1.21 | 0.39 | 0.20 | 0.15 |
| 6x700 Diagonal LSTM | 1.55 | 0.47 | 0.25 | 0.18 |
| 30x300 Gated ConvNet, 1-D conv | 1.85 | 0.47 | 0.29 | 0.20 |

## 4 RELATED WORKS

Neural networks that contain recurrent paths have been of great interest for a long time because of their ability to sequence modeling. However, the simple RNN, or so-called vanilla RNN (Elman, 1990), has the vanishing and exploding gradient problems because the recurrent path contains matrix-vector multiplication operations and the eigenvalues of the matrix cannot be easily controlled during the training (Bengio et al., 1994). The most successful RNN, so far, is the LSTM RNN, which employs the memory cells for storing the information, and the cell states are updated by the use of element-wise multiplications (Hochreiter & Schmidhuber, 1997). Since the element-wise operation corresponds to multiple independent first-order recurrent equations, and the magnitude can be easily controlled by limiting the coefficients of the recurrent equations, LSTM RNN shows fairly good trainability and long-term memorization capability. However, LSTM RNN is based on two recurrent paths, one is the cell state update and the other is the output feedback, and, as a result, the behavior is hard to interpret. Gated recurrent unit (GRU) is also a successful RNN structure; the operation also employs two different recurrences, cell state update, and output feedback to the input (Cho et al., 2014).

Recently, several studies have emerged to simplify the RNNs so that they operate in a more predictable way. Several strongly-typed RNNs are suggested in (Balduzzi & Ghifary, 2016). Strongly-typed RNNs include QRNN and SRU. The potential of QRNNs and SRUs has been explored in (Bradbury et al., 2017; Lei et al., 2017; Martin & Cundy, 2018). However, QRNNs and SRUs only

contain cell state updates and do not have direct output feedback; they are not considered to show a better performance than LSTM RNN in many applications.

Another approach to sequence modeling is using only feedforward networks, such as CNNs and Gated ConvNets (Gehring et al., 2017; Dauphin et al., 2017; Liptchinsky et al., 2017). Feedforward networks have the advantages of stable training and parallelization. However, feedforward networks only have a finite length of scope, and they tend to demand more parameters for increasing the scope.

In this work, we added 1-D convolution to increase the performance of QRNN, Gated ConvNet, and diagonal LSTM based structures. The 1-D convolution seems to supplement the lack of short-term feedback in QRNN. In addition, in Gated ConvNets, the 1-D convolution helps to increase the scope length with only a small number of parameters.

## 5 CONCLUDING REMARKS

Acoustic modeling using LSTM RNN on embedded devices is very inefficient because the model parameters need to be obtained from the DRAM at every inference, and the model size is usually much larger than that of the cache memory. This problem can be solved by computing multiple output samples at a time with one retrieval of model parameters. The so-called multi-timestep parallelization, however, cannot be applied to conventional LSTM RNN because the output is directly used for evaluation of the next one, forcing sequential computation. We explore the performances of a few acoustic models that permit multi-timestep parallelization including the diagonal LSTM, the quasi RNN (QRNN), and the Gated ConvNet. In particular, adding 1-D convolution to each layer of the networks resulted in quite good performance improvements. The QRNN and Gated Convnets augmented with 1-D convolution at each layer showed much better performances than the LSTM-based acoustic models for both end-to-end speech recognition and frame-wise phoneme classification tasks. The execution time performances on an embedded system are presented, which showed a speed-up of over 500% when the multi-timestep parallelization factor of 8 was applied. This research encourages the exploration of diverse sequence models while traditional researches have heavily relied on LSTM-based RNN.

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
