# OpenReview forum: "EXPLORATION OF EFFICIENT ON-DEVICE ACOUSTIC MODELING WITH NEURAL NETWORKS"
_ICLR.cc/2019/Conference_

### Official Review · AnonReviewer3 · 2018-11-01
**review of the paper**

**Rating:** 4
**Confidence:** 4

**Review:**

This paper investigates a number of techniques and neural network architectures for embedded acoustic modeling.  The goal is to reduce the memory access and make efficient computation, in the meantime, to sustain good ASR performance.  Overall, the paper is well motivated and well written.  However, I have following concerns.

1. It is not clear from the paper whether both the training and inference are conducted on embedded devices or only the inference?  I assume it is the latter but can't find it explicitly mentioned in the paper.

2. The exploration carried out in the paper is more on the system level and the novelty is not overwhelmingly significant.

3. My major concern is that the reported WERs on WSJ and phoneme classification accuracy are quite off.  20%-30% WERs for WSJ  do not seem to be usable in real applications.  Honestly, I don't even think this performance is better than well-trained GMM-HMM acoustic models using a Viterbi decoder.  Furthermore, there is no clear winners across the investigated architectures  in terms of performance.  One question is if one wants to deploy such an on-device system, which architecture shall be chosen?

4. A more general comment on the work explored  in the paper.  First of all, the on-device memory issue puts a heavy constraint on the capacity of acoustic models, which will significantly hurt the modeling capability for the DNN-based acoustic models.  Deep learning acoustic models can outperform GMM-HMM because they can use large model capacity with very deep and complex architectures when a large amount of training data is available.  Second, for CTC, when the training data is limited,  its performance is far worse than the hybrid DNN-HMM model, let alone a pure end-to-end fashion without using external LM and dictionary.  If WFST-based decoders (composition of WFSTs of LM, dictionary and deblank/repetition) are used, then the memory issue will surface again.

---

### Official Review · AnonReviewer1 · 2018-11-02

**Rating:** 4
**Confidence:** 4

**Review:**

This paper present a study on efficient acoustic modeling using neural networks-based model. Four approaches are presented and evaluated: diag LSTM, QRNN, Gated ConvNet and adding a 1D convolution layer. The evaluation is done on ASR task using WSJ and in phoneme classification task using the TIMIT corpus. The study show that the inference speed is improved with comparable of better performance than the standard LSTM model.

The findings presented in this paper are interesting and quite useful when one wants to implement a LSTM-based acoustic model on mobile devices. The paper is well written and easy to ready.

The main issue of this paper is the lack of novelty: the three evaluated approaches (Diag LSTM, QRNN and Gated ConvNet) are not novel, the only novelty is the addition of a 1D convolution, which is not enough for a conference like ICLR.

Minor comments on the experiments:
* The network quantization approach has been shown to lead to efficient neural networks, could the authors provide a comparison between their approach and the quantization approach ?
* On the TIMIT experiment, the authors could add a decoder and use the PER metric instead of the frame accuracy, so they could provide comparison with the literature.
* WSJ and TIMIT are quite small corpora compared to the available corpora, maybe the authors should consider using large corpora like Librispeech. It could be interesting to see the performance of the presented approaches.

Overall, this paper feels more like a technical report: the findings could be useful, but its novelty is too limited for ICLR. Hence I argue for rejection, and suggest that the authors consider submitting the paper to a speech conference like ICASSP.

---

### Official Review · AnonReviewer2 · 2018-11-03
**Review of "EXPLORATION OF EFFICIENT ON-DEVICE ACOUSTIC MODELING WITH NEURAL NETWORKS"**

**Rating:** 4
**Confidence:** 5

**Review:**

This paper discusses applications of variants of RNNs and Gated CNN to acoustic modeling in embedded speech recognition systems, and the main focus of the paper is computational (memory) efficiency when we deploy the system. The paper well describes the problem of the current LSTM, especially focusing on the recurrent connection matrix operations, which is a bottle neck in this scenario, and introduces variants of RNNs (e.g., QRNN). Also these variants may not yield enough performance compared with LSTM, but 1-D convolution and/or deep structure helps to avoid the degradation. One of the biggest issues of this paper is that they use CTC as an acoustic model, while still many real speech recognition applications and major open source (Kaldi) use hybrid HMM/DNN(TDNN, LSTM, CNN, etc.) systems. Therefore, the paper's claim on CTC is not along with the current application trends. (It may be changed near future, but still hybrid systems are dominant). For example, the WSJ WER performance listed in Table 3 is easily obtained by a simple feed-forward DNN in the hybrid system. The latest Lattice free MMI with TDNN can achieve better performance (~2.X% WER), and this decoding is quite fast compared with LSTM. The authors should consider this current situation of state-of-the-art speech recognition. Also, the techniques described in the paper are all based on existing techniques, and the paper lacks the technical novelty.

Other comments:
- in Abstract and the first part of Introduction: as I mentioned above, CTC based character-prediction modeling is not a major acoustic model.
- The paper needs some discussions about TDNN, which is a major acoustic modeling (fast and accurate) in Kaldi
- p.4 first line "and  represents element-wise multiplication": The element-wise multiplication operation was first appeared in Eq. (1), and it should be explained there.
- Section 3.2: I actually don't fully understand the claims of this experiment based on TIMIT, as it is phoneme recognition, and not directly related to the real application, which is the main target of this paper I think. My suggestion is to place these TIMIT based experiments as a preliminary experiment to investigate the variants of RNN or gated CNN before the WSJ experiments. (I did not say that Section 3.2 is useless. This analysis is actually valuable, and this suggested change about the position of this TIMIT experiment can avoid some confusion of the main target of this paper.)

---

### Meta-Review · Area_Chair1 · 2018-12-14
**Work lacks novelty and experimental validation**

**Confidence:** 5
**Recommendation:** Reject

**Metareview:**

In this work, the authors conduct experiments using variants of RNNs and Gated CNNs on a speech recognition task, motivated by the goal of reducing the computational requirements when deploying these models on mobile devices.
While this is an important concern for practical deployment of ASR systems, the main concerns expressed by the reviewers is that the work lacks novelty. Further, the authors choice to investigate CTC based systems which predict characters. These models are not state-of-the-art for ASR, and as such it is hard to judge the impact of this work on a state-of-the-art embedded ASR system. Finally, it would be beneficial to replicate results on a much larger corpus such as Librispeech or Switchboard. Based on the unanimous decision from the reviewers, the AC agrees that the work, in the present form, should be rejected.